# One-Shot Transfer of Affordance Regions? AffCorrs!

**Denis Hadjivelichkov** *
Centre for Artificial Intelligence
University College London

**Sicelukwanda Zwane**
Centre for Artificial Intelligence
University College London

**Marc Peter Deisenroth**
Centre for Artificial Intelligence
University College London

**Lourdes Agapito**
Department of Computer Science
University College London

**Dimitrios Kanoulas**
Department of Computer Science
University College London

**Abstract:** In this work, we tackle one-shot visual search of object parts. Given a single reference image of an object with annotated affordance regions, we segment semantically corresponding parts within a target scene. We propose AffCorrs, an unsupervised model that combines the properties of pre-trained DINO-ViT's image descriptors and cyclic correspondences. We use AffCorrs to find corresponding affordances both for intra- and inter-class one-shot part segmentation. This task is more difficult than supervised alternatives, but enables future work such as learning affordances via imitation and assisted teleoperation. Project page with code and dataset: `https://sites.google.com/view/affcorrs`

**Keywords:** One-Shot, Affordance, Correspondence

## 1 Introduction

Robot agents can significantly benefit from perceiving and understanding what the environment *affords* them to do. Affordances [1] are representations of how a part of the environment can be used, e.g., a spoon *affords* to be grasped, and to contain liquid. Being grounded on human semantics, they are intuitive and explainable. Part-based affordance representations can be efficiently used as an intermediate representation that reduces the dimensionality of many robot learning problems significantly [2]. Just as people are able to transfer the knowledge of an object's functionality to other objects from only a few examples, e.g., the graspable handles of jugs to mugs, it would be beneficial for robots to understand such correspondences, too. We motivate the topic of recognizing known affordance regions in unseen novel objects as a useful step toward more autonomous robots, assisted teleoperation, visual inspection, and scene understanding.

Evidently, semantic part correspondence can be achieved with fully supervised methods [3, 4]. However, they are limited to objects similar to the ones present in the dataset, each requiring multiple instances for better generalization of the object category. Meanwhile, the self-supervision and unsupervised learning paradigms, present an alternative direction, which alleviates the data annotation bottleneck [5], and could enable robots to learn continually by themselves [6].

In this work, we demonstrate how the pre-trained DINO-ViT model [7], which is shown to produce good co-segmentation and point correspondences [8], can be used for part querying and finding semantically corresponding parts in one-shot – a novel formulation of the One-Shot Instance Segmentation (OSIS) problem extended to parts, which we call One-Shot Affordance Part Segmentation (OSAPS). The method enables us to query on any segmentation mask leveraging the semantic prior given by a user, or a preceding system. In particular, we demonstrate how querying on affordance

---

*Corresponding authors are {`dennis.hadjivelichkov, dimitrios.kanoulas`}@ucl.ac.uk

6th Conference on Robot Learning (CoRL 2022), Auckland, New Zealand.

part regions can be associated with predefined skills. As far as we are aware, this work is one of the first to extend one-shot instance segmentation to part masks, and affordance parts in particular. Our proposed method presents several contributions: (i) **One-Shot** transfer of affordance regions instead of learning through supervision; (ii) **Decoupled skills** that allow the separation of affordance discovery and execution; (iii) **Query-based** transfer instead of co-part segmentation, which allows us to transfer specific semantic parts rather than parts based on visual or geometric features only; (iv) **Benchmark** subset of an affordance dataset curated for one-shot part transfer.

## 2   Related Work

Most affordance learning work focuses on fully supervised methods that learn a particular affordance, predefined in the training set as image segments, contact points or interaction regions [9, 10, 11]. Some works focus on learning from interaction combined with geometric and perceptual features [12, 13]. Model-based approaches are a potential path towards efficient knowledge transfer. However, they still rely on expensive annotations, such as full object template models [14]. One-shot affordance detection was recently demonstrated [15], but this approach is limited to instance masks and bounding boxes, which are insufficient to transfer knowledge to a robot.

Models requiring little to no supervision have great potential for robot applications that benefit from adapting quickly and learning continuously. It is of particular interest to understand how previous works in computer and robot vision solve the semantic correspondence problem in the domains of point- and part-correspondence with limited supervision.

Many works enforce the semantic relationship between two inputs through cyclic matching, either through a cyclic loss or a proxy representation. Recent examples include the use of uniform category-level representations of objects as template 3D objects [16] or unit spheres [17]. Dense Object Nets [18] extended previous work [19] and presented a method trained with self-supervision by projecting points across multiple viewpoints of the same objects, and showing how the learned object descriptors effectively generalize over other semantically similar objects. Recently, they have been extended to learn from even less data and work with more objects through optical flow of monocular videos [20], neural radiance fields [21], and unsupervised object classification [22]. Part-based methods also often use latent representations that encode each part's appearance, shape, or pose [23, 24]. Very close to our method is [25] in that it is using cycle-consistency over transformer descriptors to produce one-shot instance segmentation.

Other methods rely on large image datasets, such as ImageNet, for pre-training of vision models as means of encoding semantic and perceptual information. Such works in point correspondence [26, 27, 28, 29] and part correspondence [23, 24] show state-of-the-art performance on several computer vision benchmarks. However, these methods often struggle with highly occluded inputs with large viewpoint variance. Caron et al. [7] show that DINO-ViT, a variant of ViT [30] trained via self-supervised knowledge distillation, can produce descriptors that contain explicit information about the underlying semantic content, with properties suitable for k-nearest neighbours (kNN) search. These properties were recently used in unison with cyclic correspondence for unparalleled co-part segmentation without fine-tuning [8]. However, the co-segmentation task does not solve our problem, since it often results in segmentation that is consistent but semantically meaningless.

The literature review leads to several conclusions: (i) for any local descriptor to be suitable for semantic correspondence, it needs to be 'aware' of the full semantic context of the object that it is part of; (ii) in matching any two objects belonging to the same semantic category, not all points will always have a true correspondence, either due to occlusion or the lack or addition of parts (such as handle on a cup, or a switch on a lamp); (iii) the currently best performing models combine semantic priors provided by pre-trained models and cyclicity.

## 3   Method

The one-shot semantic instance segmentation (OSIS) problem [31] is defined as finding and segmenting a previously unseen object in a novel scene, based on a single instruction example. Similarly, in this work, we aim to solve one-shot affordance part segmentation (OSAPS): given a support reference image $I_R \in \mathbb{R}^{3 \times H_R \times W_R}$ and query mask region $M_Q \in \{0,1\}^{H_R \times W_R}$, the task is to find the

semantically corresponding region in the target image $I_T \in \mathbb{R}^{3 \times H_T \times W_T}$. The variables $H_R$, $W_R$ denote the height and width of the support, while $H_T$, $W_T$ denote those of the target, respectively.

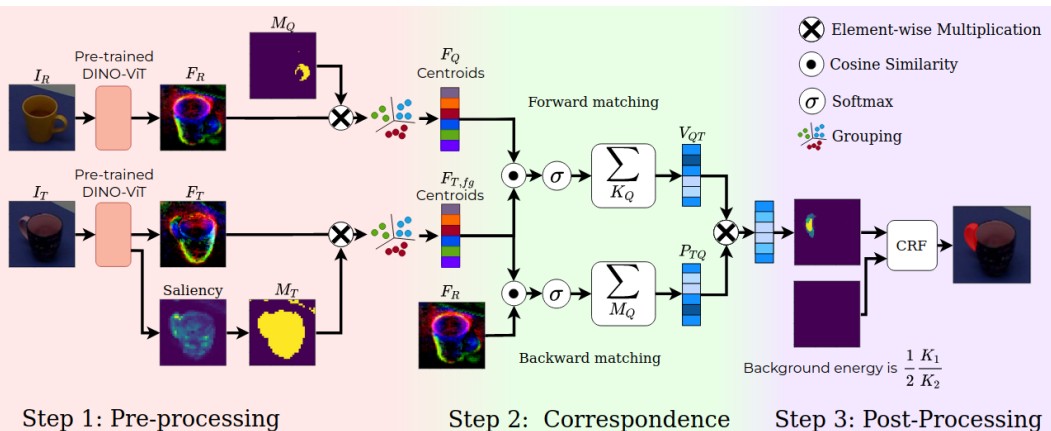

Figure 1: The overall system diagram: The support descriptors $F_R$ belonging to the query area $M_Q$ are grouped into $K_Q$ query centroids. The salient target image descriptors are grouped into $K_T$ target descriptors. Cyclic correspondence is enforced by matching $K_Q$ to $K_T$ (forward matching), and $K_T$ to $F_T$ (backward matching). A score is computed for each centroid in the target. The centroid scores are mapped to the descriptors belonging to each grouping. Finally, each pixel is determined to be foreground (or not) by the CRF, which compares the scores with a baseline energy level.

## 3.1 Query Region Correspondence

**Pre-processing:** The key takeaway from successful attempts in related works is that cyclicity plays an important role in correspondence—a matching region corresponds to the query region and vice versa (i.e., the query region corresponds to the matching region). Given the kNN-like properties of DINO-ViT, we propose to use its descriptors to find correspondences. Thus, we get descriptor images $F_R$ and $F_T$ corresponding to $I_R$ and $I_T$, respectively. Since dense descriptors retain their spatial locations, we can use the query mask $M_Q$ and target mask $M_T$ (e.g., one generated through saliency or user-provided) to only select the foreground pixels (Fig. 1-Step 1).

A conventional pixel correspondence method, such as [32], can then be applied to find matches between the two images. However, it is dealing with several issues: (i) occluded or missing parts/objects in the target (e.g., a cup that does not have a handle); (ii) some parts are more distinctive than others (e.g., the tip of a blade versus its centre); (iii) descriptors could be noisy and Euclidean distance between them is not sufficient for best-matching (see [18, 22]). We start by grouping the descriptors into $K_Q$ query groups and $K_T$ target groups (see Fig. 1-Step 2), we get mean descriptors $F_Q$ and $F_{T,fg}$, respectively, which are less noisy than single descriptors.

**Forward matching:** Through pairwise cosine similarity, we can find how close each of the $F_Q$ centroids is to each $F_{T,fg}$ centroid descriptors, as $sim(F_Q, F_{T,fg}) \in \mathbb{R}^{K_Q \times K_T}$. For each query centroid, we determine the probability that it matches the target centroids as

$$A_{QT} = \text{softmax}_{T,fg} \left( sim(F_Q, F_{T,fg}) / \tau_{QT} \right), \tag{1}$$

where $\tau_{QT}$ is the softmax temperature and $T, fg$ is the target centroid axis. We want to enable one-to-many matching, as well as matching of regions with different scales (e.g., matching a small handle to a large one). However, we find that the matching probability $A_{QT}$ in larger less distinctive areas is spread more and thus – much smaller. By summing over the query centroids as in Eq. (2), we get 'votes' $V_{QT}$ which deal with this issue by summing over the query centroids as

$$V_{QT} = \sum_Q A_{QT}. \tag{2}$$

The votes balance well between matches to distinctive areas (which are fewer but more confident) and matches to less distinctive ones (which are more numerous but less confident).

**Backward matching:** Similarly, we could find the matches from the target to the support. We match to the full support descriptor image $F_R$ rather than the centroids of the selected area $F_Q$ – with this information, any matches that correspond to another area of the support image can be excluded. Thus, our backward matching affinity is computed with the full $F_R$ as

$$A_{TR} = \text{softmax}_R \left( sim(F_{T,fg}, F_R)/\tau_{TR} \right), \tag{3}$$

while the probability that each target centroid corresponds to the query region of the support image is computed by summing over the mask query pixels $M_Q$:

$$P_{TQ} = \sum_{M_Q} A_{TR} \tag{4}$$

**Post-processing:** By multiplying $P_{TQ}$ and $V_{QT}$, we get a 'score' $S_{T,fg}$ for each target centroid, as $S_{T,fg} = V_{QT} \cdot P_{TQ}$. We map each pixel to the score value of the centroid which represents it (See Fig. 1-Step 3). Each target centroid with $P_{TQ} > 0.5$ is likely a correspondence. Meanwhile, $V_{QT}$ is not a probability, but rather a sum of $K_Q$ probabilities, each of which represents a match that is more likely than average to represent a true correspondence if it's larger than $1/K_T$. Hence, we deem the heuristic threshold $V_{QT} > K_Q/K_T$ as representative of a likely match for each target centroid. Finally, the two thresholds can be multiplied to get a 'score' threshold for each centroid.

Using a Conditional Random Field (CRF), a smooth binary mask can be produced which loosely follows an energy boundary - we set the foreground energy term to be the scores image, while the background energy – a constant of $K_Q/2K_T$ which is the score threshold.

**Design choices:** Our choice of descriptor model is the DINO-ViT-S with patch size 8, pre-trained on ImageNet, due to superior properties in the similar task of co-part segmentation [8]. The support and target descriptors filtering is done by applying the support query and saliency masks, respectively. Clustering is achieved via Fast K-Means [33] into an over-segmented image with $K_Q = K_T = 10$ [2][3]. We choose $\tau_{QT} = 0.2$, $\tau_{TR} = 0.02$ empirically. Having a large $\tau_{QT}$ means the forward matching is lenient and selects many potential candidates, while the low $\tau_{TR}$ filters in only the parts of those candidates that are confidently matching to the query part rather than the rest of the reference.

### 3.2 Affordance Transfer

We propose one-shot affordance skill transfer by defining a stack of skills $S$, i.e., affordance functions $S_i(o_{aff})$ associated with manually annotated parts (and their descriptors) $o_{aff}$ sensor inputs. Those skills would then be transferred to the corresponding parts $\hat{o}_{aff}$ of other objects by applying $S_i(\hat{o}_{aff})$. For example, a skill might be to grasp at the centroid of a graspable region, or place a ball in a containment region. In more general terms, the skill can be formulated as $S_i(\hat{o}_{aff}, o_{class}, X)$, where $o_{class}$ is the (either categorical or latent) class of the support object and $X$ is the robot's sensory state. The skills can be either robot-specific or robot-agnostic, since the chosen affordance representation does not make assumptions. In this work, we show this system with simple first-order affordances (grasping and containing) in simple scenes, and leave multi-object manipulations for the future.

## 4 Experiments

### 4.1 Quality of Part Transfer

**Metrics:** We use metrics standard in affordance learning literature [34, 35]: per-affordance class Intersection over Union (also called Jaccard index), and $F_b^w$-measure [36] which is a weighted version of IoU that accounts for pixel location and mask interpolation.

**Datasets:** As our main benchmark, we use the UMD Affordance Dataset [37] due to its variety of objects, allowing us to evaluate affordance transfer in both inter- and intra-class pairs. Similarly to how works in one-shot instance transfer use modified folds of PASCAL-5$^i$ [38] and COCO-20$^i$ [39], we present UMD$^i$ – a one-shot correspondence variant of UMD, which is composed of a single

---

[2]Any similar method such as superpixelization could work as long as the final patches are dense, loosely follow the objects' geometry and produce segments smaller or equal in size to the smallest distinctive part.

[3]We found that any value larger than 3 leads to comparable results for the used datasets. However, the number of segments should be larger than the number of distinct areas of the query parts.

instance of each object in the dataset with both RGB image and affordance ground truth annotation. The original annotations are kept to highlight the difficulties of semantic transfer, as no two human annotations are the same. The classes include common objects such as bowls, ladles, and knives, with manually labelled grasp, scoop, wrap-grasp, support, contain, cut, and pound affordances.

**Experimental Setup:** For each object in UMD$^i$, we attempt to transfer its ground truth part masks to each other object of the same class (*intra-class*) and to each other object of classes that possess the same affordance (*inter-class*) from the dataset. The quantitative metrics are averaged over each affordance type. The qualitative comparison is shown as well.

**Baselines:** As an upper baseline, we include the reported metrics of fully supervised [34, 35] on the UMD dataset (standard split for intra- and novel split for inter-class results). These are not directly comparable with our method, since they could learn the way affordances are labelled within the dataset through supervision and are evaluated on the test subset of UMD, instead of UMD$^i$. As baselines, we have included two variants of the current SotA in one-shot instance transfer–BAM [40]. We also include the unsupervised co-segmentation method [8], which inspired our work. Since this method produces 2 to 10 unsupervised segments (by using K-Means elbow point), we take all segments in the support that have at least 50% overlap with the support ground truth, and use the aggregate of their correspondences as the 'estimated parts'. This estimate is then compared with the target ground truth.

Finally, to showcase the difficulty of semantic labelling we also included a 'human level': a person is shown two objects from the same class, they are given the task of one-shot transferring labels by observing the ground truth annotation of one image and annotating the second. This is done once for each UMD$^i$ object as the target, after which the metric score is scaled by the number of objects belonging to that class before computing the per-affordance means.

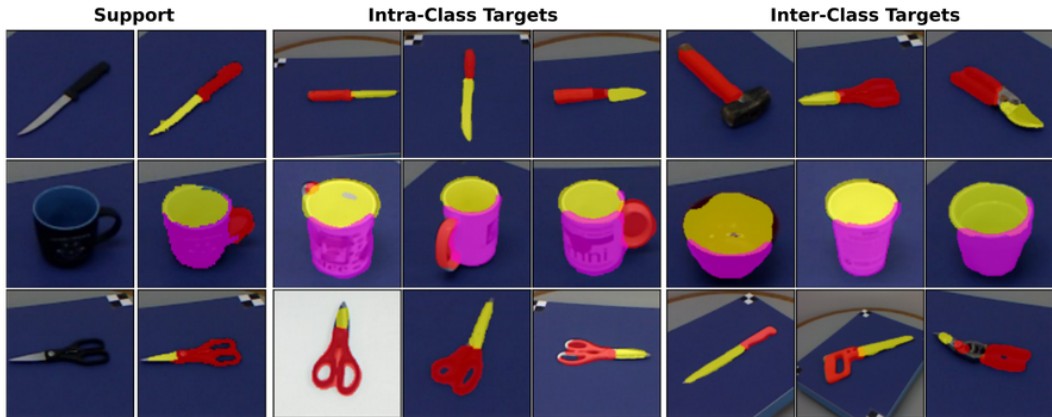

Figure 2: The annotated affordance regions from each support image are transferred with AffCorrs to intra- and inter-class targets on the same row. Colours represent grasp, cut, contain, wrap-grasp.

| | Grasp | | Cut | | Scoop | | Contain | | Wrap-grasp | | Pound | | Support | |
|---|---|---|---|---|---|---|---|---|---|---|---|---|---|---|
| | IoU | $F_\beta^w$ | IoU | $F_\beta^w$ | IoU | $F_\beta^w$ | IoU | $F_\beta^w$ | IoU | $F_\beta^w$ | IoU | $F_\beta^w$ | IoU | $F_\beta^w$ |
| Supervised | | | | | | | | | | | | | | |
| ResNet [34] | **0.71** | - | **0.79** | - | **0.86** | - | **0.86** | - | **0.84** | - | **0.72** | - | **0.55** | - |
| ADNet [35] | - | **0.73** | - | 0.72 | - | **0.80** | - | **0.85** | - | 0.81 | - | **0.87** | - | 0.76 |
| AffNet [3] | - | **0.73** | - | **0.81** | - | 0.76 | - | 0.83 | - | **0.82** | - | 0.79 | - | **0.84** |
| Unsupervised / One-Shot Transfer | | | | | | | | | | | | | | |
| BAM-ResNet [40] | 0.26 | 0.26 | 0.28 | 0.23 | 0.52 | 0.57 | 0.57 | 0.60 | 0.42 | 0.45 | 0.45 | 0.50 | 0.43 | 0.60 |
| BAM-VGG [40] | 0.15 | 0.17 | 0.17 | 0.13 | 0.43 | 0.45 | 0.56 | 0.59 | 0.41 | 0.45 | 0.39 | 0.44 | 0.27 | 0.41 |
| DINO-ViT [8] | 0.45 | 0.51 | 0.57 | 0.64 | 0.61 | 0.64 | 0.42 | 0.48 | 0.53 | 0.62 | 0.66 | 0.76 | 0.66 | 0.75 |
| **AffCorrs (ours)** | **0.55** | **0.65** | **0.72** | **0.81** | **0.73** | **0.81** | **0.82** | **0.87** | **0.83** | **0.89** | **0.78** | **0.87** | **0.82** | **0.87** |
| Human level | 0.59 | 0.79 | 0.64 | 0.82 | 0.66 | 0.83 | 0.72 | 0.79 | 0.73 | 0.74 | 0.74 | 0.74 | 0.74 | 0.75 |

Table 1: Comparison of per-affordance metrics on intra-class pairs.

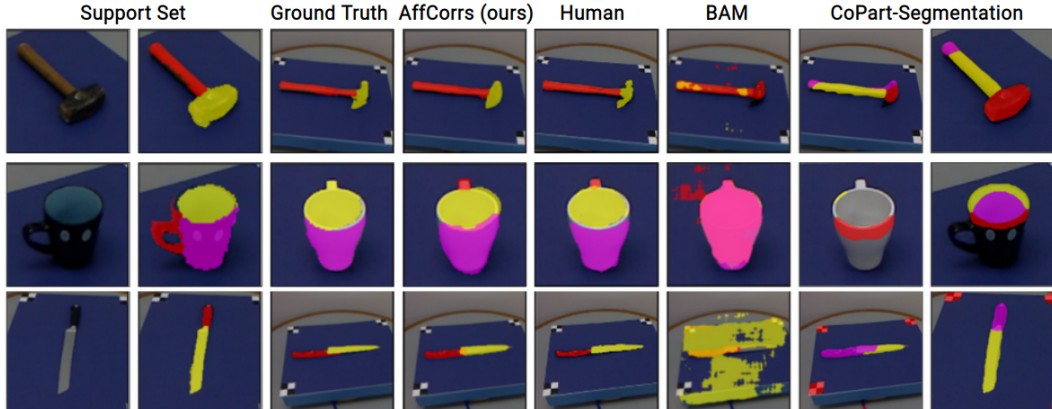

Figure 3: Visual comparison of the segmentation produced by various methods. Colours represent grasp, cut, contain, wrap-grasp. Note that Co-part segmentation has a separate colouring.

| | Grasp | | Cut | | Scoop | | Contain | | Wrap-grasp | | Pound | | Support | |
|---|---|---|---|---|---|---|---|---|---|---|---|---|---|---|
| | IoU | $F_\beta^w$ | IoU | $F_\beta^w$ | IoU | $F_\beta^w$ | IoU | $F_\beta^w$ | IoU | $F_\beta^w$ | IoU | $F_\beta^w$ | IoU | $F_\beta^w$ |
| Supervised | | | | | | | | | | | | | | |
| ResNet [34] | **0.33** | - | **0.51** | - | **0.69** | - | **0.52** | - | **0.85** | - | **0.09** | - | **0.51** | - |
| | | | | | | | | | | | | | | |
| Unsupervised / One-Shot Transfer | | | | | | | | | | | | | | |
| BAM-ResNet [40] | 0.22 | 0.25 | 0.22 | 0.25 | 0.20 | 0.21 | 0.51 | 0.54 | 0.17 | 0.18 | 0.15 | 0.16 | 0.12 | 0.13 |
| BAM-VGG [40] | 0.13 | 0.15 | 0.13 | 0.14 | 0.17 | 0.18 | 0.50 | 0.52 | 0.16 | 0.18 | 0.13 | 0.15 | 0.05 | 0.05 |
| DINO-ViT [8] | **0.39** | **0.45** | 0.50 | **0.57** | 0.58 | 0.60 | 0.30 | 0.34 | 0.56 | 0.64 | 0.66 | **0.75** | 0.68 | 0.76 |
| **AffCorrs (ours)** | **0.39** | 0.41 | **0.51** | 0.50 | **0.62** | **0.65** | **0.71** | **0.75** | **0.83** | **0.87** | **0.72** | 0.73 | **0.82** | **0.79** |

Table 2: Comparison of per-affordance metrics on inter-class pairs.

**Results:** A visualization of parts transferred with AffCorrs in Figure 2 shows that the masks are relatively robust to viewpoint variance (see mugs), and missing correspondences (see knife-to-hammer, and mug-to-cup), while being surprisingly capable of transferring affordance across dissimilar regions. In Figure 3 we show a comparison with the unsupervised baselines. Both the qualitative and quantitative comparisons (in Tables 1 and 2) affirm that AffCorrs performs better on UMD[i4] . The BAM baseline under-performs, likely due to being tailored for whole object instance transfer rather than parts. Meanwhile, the co-part segmentation, which uses the same saliency masks and backbone as AffCorrs, appears to often ignore the foreground when it doesn't deem it common enough across the objects, and produce parts that don't align with what we would consider semantically significant.

## 4.2 Affordance Transfer in the Real World

**Experimental Setup:** To showcase the application to affordance transfer and evaluate the current limitations in realistic scenes, we present the following evaluation setup: we use a Franka Emika manipulator with an arm-mounted RGB-D camera. As a query, we use a single image of a screwdriver toy with annotated grasping area, and a mug with annotated containment. The query is one-shot transferred to the robot's unseen environment. Finally, the robot attempts to use the affordance skill.

We test the method in two settings (See Figure 4) – single object and multiple objects. Single object scenes contain one object that belongs to the same class as the query (intra-class), or an object from a different class (inter-class) that has an affordance equivalent to the queried one (e.g., graspable part). In the scenes containing multiple objects, there are several objects that are 'true' correspondences (See Appx. C) , along with distractor objects. In both cases, the background was varied with different surfaces (table, carton, textile, whiteboard). The objects used are relatively spaced out, with occasional occlusion. Ten examples of either setting were used for the following robot experiments.

---

[4]The appendices detail an ablation over AffCorrs variants (Appx. A) , further outputs of AffCorrs (Appx. D, E) and a comparison with a flow-based baseline (Appx. F).

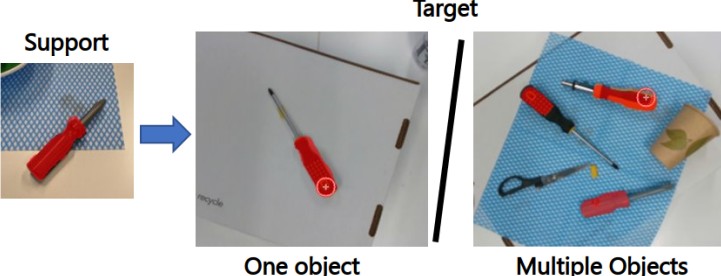

Figure 4: Examples of robot test scenes - either with one object or multiple. The red area denotes correspondences, the circle - the next object selected for manipulation based on the highest point.

Two simple first-order affordance skills are shown: grasping and containment. More complex multi-object interactions are left for future work. The grasping skill is defined as picking at the 3D centroid of a part, with grasp orientation along the largest PCA axis in XY-space (i.e., we assume a top grasp). The containment skill is defined as opening the robot gripper above the XY centroid of the affordance region. Each affordance is attempted ten times for each setting. An attempt is deemed successful if the robot successfully uses the affordance of all objects that correspond to the query, e.g., grasps all objects with tool handles *and* doesn't attempt to grasp a non-corresponding object.

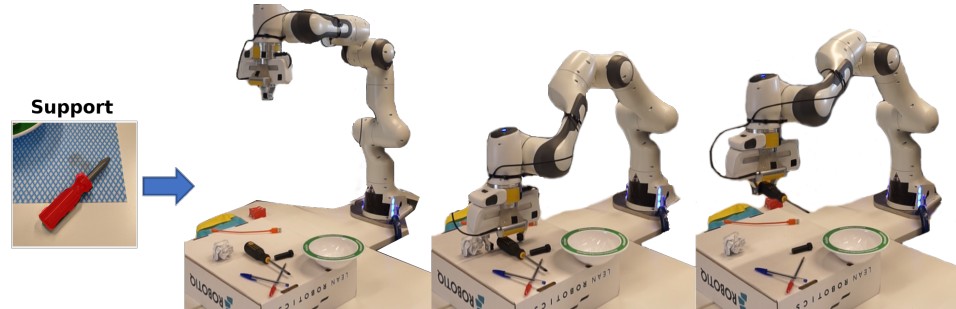

Figure 5: Example of the robot finding and using the handle-grasp affordance.

**Results:** An example grasp in multi-object scene is shown in Figure 5. In the one-object setting, we observed a 100% success rate for both grasping and containment. In the multi-object setting, the grasping success rate dropped to 70% and the containment – to 80%. Refer to Appendix B for grasping location examples and baseline comparison. As noted by previous OSIS approaches [31, 25], AffCorrs too shows that cluttered scenes are more challenging, which underscores the importance of extending the model in order to process such scenes robustly.

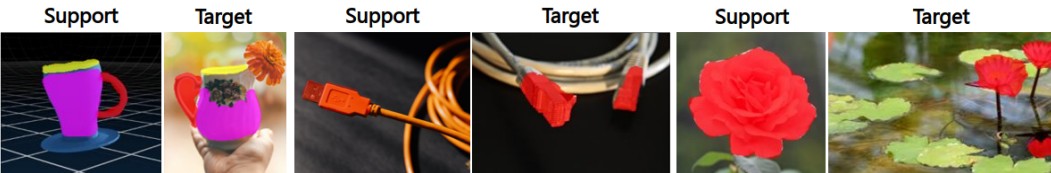

Figure 6: Several examples of one-shot transfer, from left to right: Transfer (i) from a mug simulated in Isaac Sim to a real jug; (ii) from a USB cable to an Ethernet cable; (iii) from a rose to water loti. Note that the reflection of the flower was marked as corresponding as well.

### 4.3 Transfer across Less Common Objects

Finally, we briefly motivate the practical usefulness of the one-shot property of AffCorrs over supervised methods: AffCorrs can produce good semantic part correspondences across less common objects, allowing it to work with very specific affordances that likely are not represented in any big dataset. Moreover, transferring a skill from simulation to reality could be made easy by using such an affordance representation; see examples in Figure 6.

# 5   Limitations

We highlight several important limitations of the current method (see Figure 7): AffCorrs searches for correspondences with probability-based cycle-consistency, which means that it can find correspondences between vastly different objects provided that they have similar descriptors. This is in the method's favour when the support image is suitable - e.g., when transferring from a screwdriver to a hammer's handle, but not as much when the support image is different, such as a bowl to a hammer. We have seen that the descriptor similarity alone may not be enough to always prevent incorrect matches from appearing, which limits the current model to simpler scenes without severe clutter. This is also limiting the method's ability to perform well in clutter. Meanwhile, the alternative of having a conservative matching would limit the transfer-ability across inter-class pairs.

Another issue stems from the descriptors themselves – the DINO-ViT descriptors confuse between an object's texture (e.g., a print of a dog on a mug) and an actual object (e.g., a real dog). This would mean that two same-class objects with and without textures sometimes result in different parts. The transformer's positional encoding makes the model biased toward picking a correspondence that is similarly located within the image instead of the actual correspondence. This limitation could potentially be addressed by performing flipping and cropping augmentations before the clustering steps. In terms of affordances, this work shows some simple affordance interactions, however the method could, in theory, be applied to multi-object interactions. The limitations of that direction are yet to be assessed.

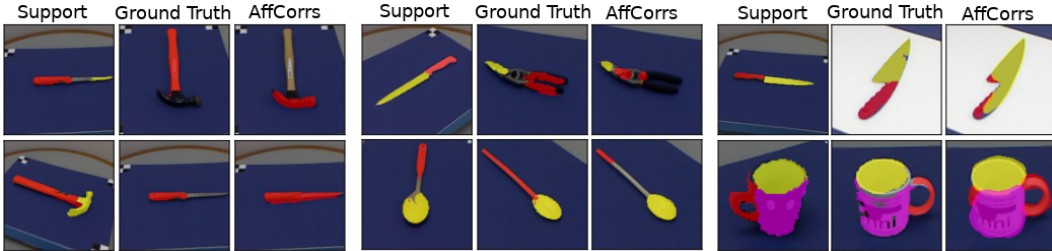

Figure 7: Six examples of model failures: some failure can likely be explained by the colour and shape of the parts shown on the top row of examples, e.g., a black knife handle mismatched to a black hammer. AffCorrs also fails due to the inherited positional encoding, observed in the bottom row examples.

# 6   Conclusion and Future Work

In this work, we showed how pre-trained DINO-ViT's descriptors can be adapted for part-based transfer of affordances. We have shown that AffCorrs is better suited for this task than the current best one-shot instance segmentation baseline and has an impressive cross-object transfer of part segmentation. However, it's limitations in working with clutter need to be addressed before any real-world application. Potential candidates to solve this problem include using an object detection model (such as DetCo [41]), more sophisticated matching, or some form of unsupervised latent classification. Solving this issue would make the method suitable for more complex dataset benchmarks, such as the IIT's affordance dataset [10]. While this work is the first tackling the one-shot affordance transfer problem, it opens the door for many future directions such as learning to both discover and use affordances from one-shot observations by looking at how people interact with objects; using affordance regions for transfer of more complex multi-object interactions; assisting teleoperation using affordance-guided object manipulation rather than hand-to-robot movement transfer.

## Acknowledgement

This work was supported by the UKRI Future Leaders Fellowship [MR/V025333/1] (RoboHike) and the CDT for Foundational Artificial Intelligence [EP/S021566/1]. For the purpose of Open Access, the author has applied a CC BY public copyright licence to any Author Accepted Manuscript version arising from this submission.

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
