# OpenReview forum: "One-Shot Transfer of Affordance Regions? AffCorrs!"
_robot-learning.org/CoRL/2022/Conference — CoRL 2022 Poster_

### Official Review · Reviewer_LmPK · 2022-07-29

**Originality:** Good
**Technical Quality:** Fair
**Clarity Of Presentation:** Good
**Impact:** 3

**Recommendation:**

Weak Reject: I recommend rejecting the paper, but will not argue for my recommendation if the majority of other reviewers have a different opinion.

**Summary:**

This work presents a novel problem of transferring part-level affordance from one labeled image to other images in the one-shot setting and proposes a one-shot transfer method that leverages pre-trained DINO-ViT image features and computes cyclic correspondence among pixels on the labeled support image and the target image. The method is quantitatively compared against two baseline methods [40] and [8], and real-world/real-robot experiments are also presented to show the part-level affordance information can be transferred from the only one given support image to other images.

**Issues:**

Please see the weaknesses.

**Quality Of The Limitations Section:**

Limitations are addressed clearly

**Reviewer Expertise:**

3: The reviewer is fairly confident that the evaluation is correct

**Robotics Focus:**

Sufficient demonstration on hardware

**Strengths And Weaknesses:**

Strengths:
- To my best knowledge, it's the first time that researchers study the problem of part-level affordance transfer in the one-shot setting. The task is very important for robotics.
- The proposed method of leveraging pre-trained image features and computing cyclic correspondence for converting pixel-level correspondence to part-level correspondence is valid, reasonable, and interesting.
- Experiments do show big improvements over the two baseline methods [40, 8], both qualitatively and quantitatively.
- Real-world/real-robot experiments are convincing.

Weaknesses:
- I'm not sure if the technical contribution of the proposed method is significant. The method directly uses and relies on the pre-trained DINO-ViT features as the per-pixel features to find the pixel-level correspondence. How about the per-pixel correspondence induced by the DINO-ViT features are not good? Then, the method mainly focuses on how to convert the pixel-level correspondence to a part-level one, by using double-way feature similarity matching (or, in another word, cyclic correspondence). The method seems quite simple to me. If the conversion from pixel-level to part-level correspondence is the only technical contribution of the proposed method, I am not sure if the technical contribution is significant enough or not.
- The authors mentioned the one-shot affordance detection work [15] in the related work but do not provide a direct comparison to it. Can you simply extract the bounding box of the part from the support image and run [15]? How would it work?
- Could you perform quantitative comparisons to the previous works that find per-pixel correspondence and directly uses the per-pixel correspondence for the part affordance on the target image, say using [18] and "Choy, Christopher B., JunYoung Gwak, Silvio Savarese, and Manmohan Chandraker. "Universal correspondence network." Advances in neural information processing systems 29 (2016)."? From these papers, it seems that the part-level correspondence can be easily obtained if the pixel-level correspondence is good enough, which seems to be the cases from checking the results presented in these papers.
- Is there any ablation study to show that the use of centroids is beneficial? The naive way is to use all the pixels.
- As shown in Table 1, why is the method performing better than humans? Is there some explanation for this? I notice some gt error for example in Fig. 3, 2nd row & third column, the handle part is wrongly labeled in GT. Is this the cause?
- The paper writing can be improved. For example, the method part is a bit hard to follow. And also, the setting for baseline methods. For instance, I don't understand what do you mean by "all parts that have at least 50% overlap" in Line 171. Why there are parts in the target image, which are overlapping with the part in the support image?

**Summary Of Recommendation:**

I like the novel and the important problem the paper proposes. However, the solution seems to be quite simple to me, since it's all about converting per-pixel correspondence to part-level correspondence using a simple cyclic aggregation. Also, some baseline comparisons need to be added.

---

### Official Review · Reviewer_oETv · 2022-07-30

**Originality:** Good
**Technical Quality:** Good
**Clarity Of Presentation:** Very Good
**Impact:** 3

**Recommendation:**

Weak Accept: I recommend accepting the paper, but will not argue for my recommendation if the majority of other reviewers have a different opinion.

**Summary:**

This work proposes a way to identify object parts in a target image given an annotated query image. The goal is to pinpoint parts that are semantically similar, e.g. given a "support" image of a knife with a segmentation mask annotating the handle & the blade, return segmentation masks for handles and blades of knives with a different shape or orientation in the target images. Authors also demonstrate success for inter-class examples, such as obtaining a region for a handle of a hammer given the support image with a mask for a handle of a knife.
The work employs pre-trained transformer models from the literature for pre-processing, then applies a novel way to impose correspondence between semantically similar parts of the objects. The approach is conceptually similar to some of the prior works that imposed cycle consistency with a use of a segmentation mask [citation 25], but the proposed approach includes a clustering step and a soft way to match clusters in query and target images.

**Issues:**

As of now, the robot experiments are limited to a very simple "from the top" grasp strategy and just two sub-tasks: a simple pickup by a long flat handle area, and a drop into a containment area. It should be possible to either attempt a wider range of grasping strategies or expand the set of tasks. For example, grasping cup handles from various angles; or completing a set of tasks after grasping. It would be particularly interesting to focus on inter-class examples, since the most relevant case for robotics would be to avoid having to obtain an appropriate support image for every object class that a robot might encounter in its environment.

**Quality Of The Limitations Section:**

Limitations are addressed clearly

**Reviewer Expertise:**

4: The reviewer is confident but not absolutely certain that the evaluation is correct

**Robotics Focus:**

Sufficient demonstration on hardware

**Strengths And Weaknesses:**

Strengths:
The abstract, introduction and related work sections are well written and clear.
The problem formulation and contributions are also stated clearly, which makes it easy to understand the scope of the paper.
Quantitative experiments on image datasets (annotated with segmentations and affordances) show a non-negligible improvement over previous approaches.
The proof-of-concept experiments with grasping using a real robot show author's commitment to make the work relevant to robotics, instead of limiting evaluation to only the static image datasets.

Weaknesses:
For this to be a strong contribution to the robotics community, the hardware experiments section should be significantly expanded.
This work presents results with a fixed grasping policy, which should be very quick to execute.
Hence, it should be easy to present comparisons with grasping that use at least one other baseline method (e.g. methods in tables 1 & 2). This would elucidate the impact of the improved detection of affordance regions on the target task of grasping. It could be the case that with a simple grasping policy the improved precision on static images does not translate to a significantly improved success rate for grasping (or alternatively, it could in fact have a visibly positive impact on grasp success). It would also be good to include experiments on a wider range of objects and tasks. As of now, the robot experiments are limited to just two sub-tasks: a simple pickup using a long flat handle area, and a drop into a containment area.


**Summary Of Recommendation:**

From the robotics perspective: this work has the potential of being useful for task-oriented grasping. Experimental section shows visualizations with qualitative results, quantitative results that compare to several previous works on segmentation, and proof-of-concept experiments robot experiments with grasping. This shows that the work is promising, but to clearly demonstrate its potential, it would be necessary to expand the set of real robot experiments.

The "Limitations" section shows a number of clear failures of the method and gives an intuitive explanation of why the method fails in this particular way. The section seems to be honest and doesn't shy away from showing very visible failures, which would be useful for others in the community who might want to build on this work. Hence, this clear "Limitations" section strengthens the presentation of this work by providing intuition for the limitations of the proposed method.

---

### Official Review · Reviewer_Mo7Z · 2022-08-01

**Originality:** Very Good
**Technical Quality:** Very Good
**Clarity Of Presentation:** Very Good
**Impact:** 4

**Recommendation:**

Strong Accept: I recommend accepting the paper and will argue for my recommendation even if other reviewers hold a different opinion.

**Summary:**

This work tackles one-shot visual search of object parts. Given a reference image with a query area marked using a mask, the AffCorrs model can segment the parts of the target image that correspond with the reference image's query area. The method first uses a pretrained DINO-ViT to generate image descriptors for the reference and target images. The descriptors are then masked and clustered, and corresponce between reference and target image descriptors is computed using cyclic correspondence. A conditional random field is used to determine whether pixels are foreground or background. Results are shown on a variant of the UMD dataset, and the authors show that their method outperforms the BAM and co-part segmentation baselines. They benchmark on both inter-class and intra-class image pairs. They also show that their method works in a real robot tabletop pick and place setup.

**Issues:**

- I found Sec. 3.1 somewhat hard to follow. Perhaps a figure illustrating just the query correspondence in greater detail might help?

**Quality Of The Limitations Section:**

Limitations are addressed clearly

**Reviewer Expertise:**

4: The reviewer is confident but not absolutely certain that the evaluation is correct

**Robotics Focus:**

Sufficient demonstration on hardware

**Strengths And Weaknesses:**

Strengths:
- One-shot transfer with an off-the-shelf pretrained model rather than supervised learning is very appealing, as supervised learning typically requires collecting a large custom dataset for each new setup
- The method gives the user flexibility to specify different parts of objects for different end effectors. This is important as affordances varies with different end effectors and different tasks.

Weaknesses:
- While AffCorrs is demonstrated in a 2D pick-and-place setting, the 2D representation of affordances might not work well for robotic tasks that require 6-DOF, such as approaching a mug from the side to grasp it by its handle
- Relatedly, the centroid of a segmented part is not always the best place to grasp an object


**Summary Of Recommendation:**

The paper proposes an interesting way to leverage pretrained vision models for affordance prediction in pick-and-place tasks. The method is elegant, and the experiments are well-executed, showing superior performance over reasonable baselines. While the applicability of the method may be somewhat limited for tasks that are not 2D pick-and-place, I believe this paper should very much be of interest and usefulness to the community.

---

### Official Review · Reviewer_Zz3M · 2022-08-02

**Originality:** Good
**Technical Quality:** Good
**Clarity Of Presentation:** Good
**Impact:** 3

**Recommendation:**

Weak Reject: I recommend rejecting the paper, but will not argue for my recommendation if the majority of other reviewers have a different opinion.

**Summary:**

This work tackles the task of transferring an annotated region of interest (e.g. handle) from an image of object to an image of a different (but similar) object. In the specific use-cases considered, these regions typically correspond to possible interaction targets (although the approach is agnostic to this).

The overall framework consists of leveraging meaningful pre-trained visual features for image patches (using DINO-ViT) and using these to find correspondences between the annotated ‘query’ region in the reference image and the foreground regions in the target image. The key idea is to first cluster the features in the source and target image, and identify clusters of the target features that are both: a) mapped from query pixels when computing source -> target correspondences, and b) are mapped to query pixels when computing target -> source correspondence.

The empirical results show this leads to better correspondences than similar unsupervised baselines.


**Issues:**

Adding the requested ablations/baselines would be helpful.

I would also be curious to hear the authors’ opinion on why they view this work as tackling affordance transfer, rather than more generally ‘region transfer’ (and evaluate it as such)?


**Quality Of The Limitations Section:**

Limitations are addressed clearly

**Reviewer Expertise:**

4: The reviewer is confident but not absolutely certain that the evaluation is correct

**Robotics Focus:**

Relevant but unlikely to deploy to hardware in near future

**Strengths And Weaknesses:**

Strengths:
- On the plus side, the overall approach is simple and intuitive. It uses strong unsupervised features to find correspondences between a query region in a source image to a target images, and several of the design choices e.g. using clusters, smoothing with CRFs, combining correspondence scores in both directions etc. make intuitive sense.

- The empirical results show that the proposed method does outperform a prior work which uses similar features, but instead predicts (corresponding) unsupervised parts.

- The paper also shows some results on transfer across categories and the results are encouraging.

Weaknesses:
- The approach presented here is a rather generic one, and I am not sure why the work is positioned as focusing on affordances — nothing in the method actually relies on the regions being affordance. In fact, the term ‘affordance’ may generally also subsume the mode of interaction (e.g. how the action should be performed) and this work certainly does not predict transfer of this (e.g. 6D relative orientation of hand). In fact, this paper presents a method for transferring annotated masks, and actually should have been presented as such.

- While there are some reported baselines, I feel the text is missing a few rather informative ablations/baselines. In particular:
a) Baseline 1: what happens is one computes correspondence via naive nearest neighbors using DINO features?

b) Baseline 2: The vision community has previous had the observation that deep features are good for correspondence. For example [A] showed that instead of naive nearest neighbors, using deep features + flow lead to better correspondences. It would be helpful to know whether the proposed approach performs better than this baseline (modified to use DINO features).
[A] Do Convnets Learn Correspondence? Long et. al.

c) Ablation: Is the combination of V_{QT} and P_{TQ} necessary? What happens if one uses just one of these terms (with an appropriately chose threshold)?


**Summary Of Recommendation:**

Overall, I think the paper tackles a useful task and proposes a simple but elegant approach. However, I am not convinced that this should be framed as a ‘affordance transfer’ work as nothing in the method is specific to this fact. On a related note, I think more baselines/ablations which convince the reader that this is indeed a good way for transferring annotated regions are required.

---

### Meta-Review · Area_Chair_rdt9 · 2022-08-10

**Recommendation:** Accept (Poster)
**Confidence:** 4

**Metareview:**

Reviewers commended the proposed method for being simple and reasonable, while being surprisingly effective.  The results show that the proposed method shows significant improvement over previous work.  Reviewers also appreciated the real robot experiments, especially the additional experiments added during the rebuttal.  Reviewers felt that this paper could provide a useful method for the community to see and build upon. The additional experiments added in the appendix (A through D) are all greatly appreciated by the reviewers. These additional experiments should be added (or at least referred to) in the main text of the final version.

Some reviewers felt that the task was inappropriately named (“affordance region transfer” vs “segmentation/mask/region transfer”, without the word “affordance”).  Perhaps this issue could be most easily resolved by clarifying at the beginning of the paper that the proposed method could be used to transfer any type of segmentation mask, but in this work the focus of the experiments is on transferring affordance regions (and evaluating on robot grasping experiments).

The paper is also missing a comparison to some simple baselines, such as the one suggested by reviewer Zz3M (DiNO features + flow). Regarding the reviewer comment: “Warping-based solutions … do not solve the one-to-many correspondence problem” - this can be dealt with by computing the flow from the target image back to the source image. Adding such a baseline would greatly strengthen the paper.

A final criticism by the reviewers was that the proposed method is somewhat limited in technical novelty.  On the other hand, the fact that such a simple method outperforms the baselines is a strength of the proposed work.